# Social Interaction and Life Satisfaction among Older Adults by Age Group

**DOI:** 10.3390/healthcare11222951

**Published:** 2023-11-12

**Authors:** Jeong-Hye Park, Se-Won Kang

**Affiliations:** 1Department of Nursing, Gyeongsang National University, Dongjin-ro 33, Jinju-si 52725, Republic of Korea; masternur@gnu.ac.kr; 2Department of Nursing, Dongseo University, 47 Jurye-ro, Sasang-gu, Busan 47011, Republic of Korea

**Keywords:** social interactions, life satisfaction, older adults, elderly

## Abstract

This study examined the effect of social interaction on life satisfaction in older adults. A total of 8188 participants were selected according to the inclusion criteria. SPSS Windows software (version 23.0) was used for statistical analyses. Data were analyzed using the independent samples *t*-test, chi-squared test, and multiple regression analysis with weights based on two age groups: 65–74 and ≥75 years. The results showed that social interaction factors influenced the life satisfaction of older adults. In the 65–74 age group, factors that statistically significantly increased life satisfaction were meeting children (β = 0.17, *p* < 0.001) and volunteer activities (β = 0.04, *p* = 0.007). In the ≥75 age group, factors that significantly increased life satisfaction were talking with friends (β = 0.11, *p* = 0.002), talking with children (β = 0.07, *p* = 0.013), using senior citizen community centers (β = 0.08, *p* = 0.001), and hobby club activities (β = 0.07, *p* = 0.001). In order to increase the life satisfaction of older adults, different ways to support social interactions need to be explored.

## 1. Introduction

South Korea is expected to become a super-aged society, in which the proportion of the population aged 65 years or older will exceed 20% by 2025. As the median age surpasses the mid-40s and life expectancy increases, the older adult population is expected to grow faster than in any other country [1]. Considering these demographic changes, it is essential to promote a happy and satisfying life in old age.

Life satisfaction refers to an individual’s cognitive and emotional evaluation of their life [2]. High life satisfaction means that individuals perceive their lives as meaningful, are hopeful, and are experiencing happiness [3,4]. High life satisfaction in old age indicates that individuals evaluate their lives positively and adapt well to the challenges of aging [5]. Previous studies have revealed various factors related to life satisfaction in old age, including living environment, interactions with neighbors or children, leisure activities, subjective health status, economic well-being, and social relationships [6]. 

Social interaction is a crucial predictor of life satisfaction [7,8]. Social interaction encompasses an individual’s relationships and interactions with others [9], including various aspects such as the individual’s network of relationships, activities, and the frequency of contact with the people they interact with [10]. Social interactions can provide psychological support and reduce social isolation, loneliness, and depression [10,11]. Psychological support from close individuals such as family members, friends, and neighbors has been reported to have a positive impact on life satisfaction [12]. Even in old age, it is vital to actively engage in social networks, maintain contact with people, receive psychological support from those close to us, and experience satisfaction with relationships. However, as individuals age, their physical function tends to deteriorate, leading to decreased activity levels. Consequently, social interactions often decline over time and are influenced by factors such as children gaining independence, retirement, grief, or losing close friends or loved ones [13]. Maintenance of social interactions may surpass these life stages [14,15]. In particular, the number of older people living alone or with only a spouse has increased rapidly. These individuals tend to have less frequent intergenerational contact than those living with their children, making them more susceptible to social isolation and loneliness [16,17,18].

Life satisfaction among older adults is a complex and multifaceted phenomenon [19]. Numerous factors contribute to the overall life satisfaction of older adults. Notably, social connections and relationships play a crucial role in older adults’ life satisfaction, with strong family ties and supportive social networks associated with higher levels of happiness [20]. Furthermore, physical health and access to quality medical services directly affect an individual’s ability to enjoy daily activities and maintain independence, making them vital factors for life satisfaction in old age [21]. Financial stability and security are also essential because financial difficulties can lead to stress and reduced life satisfaction among older adults [22]. Additionally, engaging in meaningful activities, such as finding a purpose, volunteering, and pursuing hobbies, increases life satisfaction among older adults [23]. In summary, comprehending and addressing these factors can enhance the quality of life of older adults.

Meanwhile, as the average life span has been extended due to the development of medical technology, it has been suggested that there is a need to subdivide the life cycle of the elderly. The life cycle of older adults can be categorized into the young-old (65–74), old-old (75–84), and oldest-old (85 years or over) [24]. Alternatively, it can be classified as young-old (65–74) and old-old (75 years or older) [25,26]. Young-old people are younger, relatively healthy, and lead independent lives, while old-old people experience impairment of physical and mental functions, resulting in increased health and living dependency [26]. In terms of life satisfaction, young-old people appear to be healthier and more satisfied than old-old people; however, young-old people may have lower self-esteem due to a reduction in social roles [24]. In addition, although young-old people tend to adapt better to their new roles, old-old people are more vulnerable than young-old people to problems such as deteriorating physical health and experiencing serious life events. As a result, old-old people may have many difficulties adjusting to life in old age [25,26]. Even at the present time, when the issue of the elderly is becoming visible as a social problem, detailed studies conducted on old age are still lacking.

Although various studies have been conducted on the relationship between social activities and the life satisfaction of the elderly, there is a lack of studies that reflect the detailed influence of social activities on life satisfaction or the characteristics of the elderly age group. Social interaction is an activity that reflects individual characteristics, making it difficult to interpret the results as a small-scale research study. Therefore, if possible, large-scale population surveys are required. Based on these research results, it is necessary to suggest a direction to increase the life satisfaction of the elderly and to establish detailed social exchange service policies and systems suitable for the early and late years of old age.

This study investigated the life satisfaction of older individuals with a specific focus on social interactions. To achieve this, it is imperative to establish a clear understanding of the relationship between social interactions and life satisfaction, especially by age group. Our study involved a comprehensive review of previous research encompassing various facets of social interaction in the context of aging. This included examining the social activities within older adults’ social networks and assessing the frequency of their contact with children and friends. We analyzed these factors by age group to identify how social interactions influenced life satisfaction across different stages of aging.

### Study Purpose

This study aimed to identify the effects of social interactions on life satisfaction by age group. The specific purposes are as follows:(1)To examine differences in life satisfaction and social interaction by age group.(2)To identify social interaction factors that influence life satisfaction by age group.

## 2. Materials and Methods

### 2.1. Study Design

This study used secondary data analysis to investigate the relationship between social interaction and life satisfaction among older adults.

### 2.2. Study Participants and Sampling

The study focused on individuals aged 65 years or older, selecting 8188 participants from the 2020 Elderly Survey based on the following criteria: (1) those who responded to the survey independently without input from their children or others, and (2) individuals living alone or only with a spouse, totaling one or two residents.

Individuals aged 65 years or older were divided into two groups: 65–74 years and ≥75 years according to the criteria of Neugarten, Moom, and Low [25]. Of the 8188 individuals, 4942 people were in the early geriatric phase and 3246 were in the late geriatric phase.

Data for this study were obtained from the 2020 Survey on the Status of the Elderly conducted jointly by the Ministry of Health and Welfare and the Korea Institute for Health and Social Affairs. This triennial survey aimed to assess older adults’ current circumstances and characteristics to inform welfare policies and enhance the quality of life for this growing demographic. Data collection involved one-on-one interviews conducted by 169 trained interviewers, using a questionnaire developed by the research team. Interviews were conducted between September 14 and November 2020. The survey sample design first involved stratification of Korea’s 17 cities and provinces, with the nine provinces further stratified into eastern areas, counties, and towns. The sample size was approximately 10,000 individuals. A two-stage cluster sampling method was employed to prevent under-sampling in areas with smaller populations, thereby ensuring a representative sample. The initial raw data included 10,097 participants; 8188 were selected based on the criteria for participant selection.

### 2.3. Study Variables

#### 2.3.1. Social Interaction

Social interaction was divided into two sections: (1) social activities, and (2) social contact, based on a review of previous studies [11], and the following variables were included: (1)Social Activities

Social activities included hobby clubs, social gatherings, religious activities, volunteer activities, and the use of senior citizen community centers. Social activities were measured based on the frequency of each activity.

The score of hobby clubs, social gatherings, religious activities, and volunteer activities refers to the frequency of activities at weekly or monthly intervals and is measured from 0 to 5, with higher scores indicating more frequent activities (0 = none; 1 = less than once a month; 2 = once a month; 3 = less than once a week; 4 = once a week; and 5 = more than twice a week). Hobby clubs refer to favorite or recreational activities. Social gatherings are for social purposes such as class reunions. Religious activities refer to meetings or group activities. Volunteer activities refer to individual or group activities. The use of senior citizen community centers was measured according to the frequency of use per week. The higher the number, the more frequent the use (0 = none; 1 = once a week; 2 = twice a week; 3 = three times a week; 4 = four times a week; 5 = more than five times a week).

Community centers for senior citizens are public welfare facilities for the elderly where one can chat with seniors of a similar age, participate in various programs for the elderly, and take advantage of free meals provided by the facility.

(2)Social Contacts

Social contact was measured based on the frequency of meetings and phone calls to children and friends. Frequency of contact was measured with a score from 0 to 5, with higher scores indicating more frequent contact (0 = none; 1 = once or twice a year; 2 = once or twice a quarter; 3 = once or twice a month; 4 = once a week; 5 = more than twice a week).

#### 2.3.2. Life Satisfaction

Life satisfaction refers to an individual’s level of content or happiness in their present life. Life satisfaction was measured on a five-point Likert scale ranging from 1 (not satisfied at all) to 5 (very satisfied) with a single question: “How satisfied are you with your current life generally?”

### 2.4. Adjusting Variables

The following variables were adjusted by reviewing previous studies: sex, education level, religion, living arrangement, residential area, independence in daily life, perceived health status, perceived health status of spouse, housing ownership, annual household income, number of surviving children and grandchildren, number of intimate kinships, and number of friends.

### 2.5. Ethical Considerations

The data used in this study were obtained from publicly available sources accessible on the MicroData Integration Service homepage [27]. The data were analyzed with approval from the Ethics Committee of the Korea Institute for Health and Social Affairs (IRB No: 2020–36). Notably, the study ensured the participants’ privacy, as no personal identification information was available or used, making it impossible to identify individual participants.

### 2.6. Data Analysis

Data were analyzed using the IBM SPSS Statistics version 23.0 program (IBM, Armonk, NY, USA).

(1)Participants’ general characteristics are presented as frequencies, percentages, means, and standard deviations by age group. Differences in general characteristics by age group were analyzed using the chi-square test and independent samples *t*-test.(2)Differences in life satisfaction and social interactions by age group were analyzed using the mean, standard deviation, and independent sample *t*-tests.(3)To identify the social interaction factors related to life satisfaction by age group, multiple regression analysis was performed with weights applied for parameter estimation. Before regression analysis, the suitability of the regression model was confirmed.(4)The significance level was set at 0.05.

## 3. Results

### 3.1. Differences in General Characteristics of Subjects by Age Group

Among the 8188 subjects, 4942 (60.4%) were aged 65–74 years and 3246 (39.6%) were over 75 years old. Table 1 shows the differences in participants’ general characteristics according to age group. 

In the 65–74 age group, 60.7% were women and 39.3% were men. Regarding education level, 65.6% of the participants had received up to secondary education and 59.8% followed a religion. Living with a spouse only accounted for 69.9% of the sample, and those living in rural areas accounted for the majority (41.6%). Overall, 94.7% of the cases could live independently, the perceived health status was 3.5 (±0.80) points out of 5 points, and the spouse’s perceived health status was 3.6 (±0.78) points. A total of 81.5% owned a house, and the annual average household income was 2771.3 (±3592.17) million won. The average number of children was 2.4 (±1.08), the number of grandchildren was 3.5 (±2.38), the number of intimate kinships was 2.3 (±1.95), and the number of friends was 3.2 (±2.76).

In the age group of 75 years or older, 60.3% were women, and 69.0 had received elementary education. In total, 57.4% had a religion. Furthermore, 50.2% were single-person households, 58.5% were rural area residents, and 83.1% lived independently in daily life. Perceived health status was 3.0 (±0.89) points out of 5 points, and perceived spouse health status was 3.2 (±0.89) points. The average annual household income was 1523.1 (±1668.90) million won, 76.8% owned a house, the average number of children was 3.4 (±1.50), the number of grandchildren was 5.6 (±3.32), the number of intimate kinships was 1.7 (±1.82), and the number of friends was 2.8 (±2.32). 

There was no sex difference between the two age groups (χ^2^ = 0.12, *p* = 0.729). Nonetheless, there was a significant difference in all other variables, and the variable showing the most crucial difference was education level (χ^2^ = 1343.10, *p* < 0.001). 

### 3.2. Differences in Life Satisfaction and Social Interaction by Age Group

The differences in life satisfaction and social interaction according to age group are shown in Table 2.

The frequency of all social activities showed statistically significant differences between the two groups. The activity that showed the greatest difference was social gatherings (*t* = 28.69, *p* < 0.001), and the frequency of the activity was significantly higher in the 65–74 age group. The activity that showed the next largest difference was using senior citizen community centers (*t* = −25.17, *p* < 0.001), and the frequency was significantly higher in the ≥75 age group. In terms of social contacts, the variable that showed a statistically significant difference between the two groups was meeting friends (*t* = −4.91, *p* < 0.001), and the frequency was significantly higher in the ≥75 age group.

### 3.3. Social Interaction Factors Affecting Life Satisfaction by Age Group

The social interaction factors associated with life satisfaction by age group are shown in Table 3.

In the 65–74 age group, factors that statistically significantly increased life satisfaction were meeting children (β = 0.17, *p* < 0.001) and volunteering activities (β = 0.04, *p* = 0.007).

In the ≥75 age group, factors that statistically significantly increased life satisfaction were talking with friends (β = 0.11, *p* = 0.002), talking with children (β = 0.07, *p* = 0.013), using senior citizen community centers (β = 0.08, *p* = 0.001), and hobby club activities (β = 0.07, *p* = 0.001).

## 4. Discussion

This study identified social interaction factors that affect life satisfaction by age group. 

### 4.1. Differences in Life Satisfaction and Social Interaction by Age Group

The life satisfaction score for the two age groups was 3.6 for the 65–74 age group and 3.4 (out of 5) for the ≥75 age group, showing that both age groups were above average. It was also higher in the 65–74 age group. These results show that when comparing the elderly by segmentation, the increase in chronic physical diseases and the decline in activities of daily living are accelerated in the 75 years or older (old-old) group and have a similar negative impact on life satisfaction [28,29]. In addition, with increasing age, not only does the socioeconomic situation, such as income, assets, and loss of spouse, worsen, but depression also increases and life satisfaction decreases [30,31].

Additionally, when comparing the social interaction of the two age groups in terms of the frequency of social activities, most activity items were higher in the 65–74 age group. Hobby clubs, social gatherings, religious activities, and volunteer activities were frequent in the 65–74 age group, and only the use of senior citizen community centers was common in the ≥75 age group. The 65–74 age group is early senior citizens and is likely to have a better level of physical function than the 75 or older age group. According to the 2020 Disability Survey [32], the proportion of people aged 65 or older among all disabled people was 49.9%, of which 22% were aged 65 to 74 and 27.9% were aged 75 or older, showing a low rate of disability among the elderly in early old age.

In old age, cognitive function and visual and perceptual abilities decline, and the level of self-perceived hearing declines rapidly between early and middle age [22]. In visual acuity and hearing tests, those under 75 years of age were more likely to experience a decline than those over 75 years of age. This method was reported to be superior [23]. Early old age is a period when physical functions are good enough to feel that one is still young enough to work, and efforts are made to maintain social relationships and roles [25]. Due to the characteristics of these age groups, the subjects of this study also had a high frequency of club activities for hobbies, social gatherings, religious activities, and voluntary activities requiring external physical activity in the 65–74 age group, and stable activity in the ≥75 age group. It was found that the frequency of use of senior citizen community centers, which mainly consist of indoor activities, is high.

### 4.2. Social Interaction Factors Affecting Life Satisfaction by Age Group

The study demonstrated that social interaction plays a significant role in determining the level of life satisfaction among different age groups. Notably, meeting children (*p*< 0.001) and volunteer activities (*p* = 0.006) were identified as important factors for individuals aged 65–74. For those aged ≥75, influential factors were phone calls with friends (*p* = 0.002), phone calls with children (*p* = 0.013), utilization of senior citizen community centers, and participation in hobby-based club activities (*p* = 0.001). In light of the study’s findings, the socioemotional selectivity theory [33] may have an impact on life satisfaction through social interactions in older adulthood. The theory posits that as people age, they invest more in activities and resources that serve emotionally meaningful objectives. In essence, one’s sources of motivation and goals vary according to the passage of time. Young adults hold an optimistic stance toward the future and value knowledge-related objectives as crucial and lasting goals. As such, individuals with a perception of extended future horizons tend to gravitate towards novelty-seeking experiences that satisfy their intellectual curiosity, rather than emotional gratification [34]. In contrast, middle-aged and older individuals possess a restricted view of the future as they feel that ‘there is not much life left.’ Therefore, they prioritize short-term emotional goals and direct their attention toward positive stimuli instead of negative stimuli. This translates to middle-aged and older individuals preferring to attain emotional well-being through contentment in amicable and pleasant relationships, such as those with family, relatives, and friends [35]. The study indicated that life satisfaction was primarily impacted by visiting children and friends, making phone calls, and other related factors.

Furthermore, volunteer work, utilizing senior citizen community centers, and pursuing hobbies through club activities were found to be significant variables relating to social–emotional choice theory and the pursuit of emotional stability amongst elderly individuals.

Through the results of this study, we can think about the following. First, this study can serve as a foundational study in studying the contents of activities by age group and the happy and satisfied lives of the elderly in future. Second, opportunities for various personal and social activities should be provided so that the social interactions of the elderly can continue without interruption. For the elderly population whose level of education is increasing, it is necessary to promote various social participation activities of the elderly based on practical talent donation beyond leisure and social activities. Third, a social culture of retirement preparation for the elderly is needed. We suggest that government support and community efforts are needed to ensure that individuals continue to maintain their social capabilities even after retirement and lead a satisfying life.

### 4.3. Study Limitations

This cross-sectional study used secondary data and has several limitations. First, because existing data were used, there were limitations in setting variables related to social interaction. Second, as this was a cross-sectional study, there are limitations in inferring causal relationships between the variables over time. Future studies that address these limitations, such as adding variables related to social interaction satisfaction and conducting longitudinal studies, are needed.

## 5. Conclusions

Old age refers to the period when social relationships and interactions decrease because of physical aging and retirement. However, in the 65–74 age group, participating in social activities, volunteering, and meeting children or friends on a regular basis can significantly boost life satisfaction. For those aged 75 or older, calling children and friends on the phone, utilizing senior citizen community centers, and engaging in club activities for hobbies can have a significant impact on life satisfaction. These findings show that meaningful social interaction could be more and more necessary as we approach the latter stages of old age, not frequent or active social interactions. Therefore, to enhance the quality of life for older adults, it is crucial to comprehend their social interactions accurately.

## Figures and Tables

**Table 1 healthcare-11-02951-t001:** Differences in general characteristics of subjects by age group.

Variables		65–74 Age Group (n = 4942)	≥75 Age Group (n = 3246)	χ^2^/*t*(*p*)
		N (%) or Mean ± SD		
Sex	Men	1942 (39.3)	1288 (39.7)	0.12 (0.729)
	Women	3000 (60.7)	1958 (60.3)	
Age (years)		69.0 ± 2.79	80.1 ± 4.01	−136.78 (<0.001)
Educational level	≤Elementary education	1381 (27.9)	2241 (69.0)	1343.10 (<0.001)
	Secondary education	3241 (65.6)	898 (27.7)	
	≥Higher education	320 (6.5)	107 (3.3)	
Religion	Do not have	1988 (40.2)	1382 (42.6)	4.46 (0.035)
	Have	2954 (59.8)	1864 (57.4)	
Living arrangement	Living alone	1487 (30.1)	1630 (50.2)	336.62 (<0.001)
	Living with spouse only	3455 (69.9)	1616 (49.8)	
Residential area	Capital area	1371 (27.7)	452 (13.9)	291.08 (<0.001)
	Big cities	1516 (30.7)	894 (27.5)	
	Rural area	2055 (41.6)	1900 (58.5)	
Independence in daily life	Do not need	4682 (94.7)	2698 (83.1)	297.49 (<0.001)
	Need	260 (5.3)	548 (16.9)	
Perceived health status (Range: 1–5)		3.5 ± 0.80	3.0 ± 0.89	27.43 (<0.001)
Perceived health status of spouse (Range: 1–5)		3.6 ± 0.78	3.2 ± 0.89	16.72 (<0.001)
Housing ownership	Own one’s house	4026 (81.5)	2492 (76.8)	26.58 (<0.001)
	Other	916 (18.5)	754 (23.2)	
Annual household income *		2771.3 ± 3592.17	1523.1 ± 1668.90	21.19 (<0.001)
Number of surviving children		2.4 ± 1.08	3.4 ± 1.50	−31.72 (<0.001)
Number of surviving grandchildren		3.5 ± 2.38	5.6 ± 3.32	−30.15 (<0.001)
Number of intimate kinships		2.3 ± 1.95	1.7 ± 1.82	14.63 (<0.001)
Number of friends		3.2 ± 2.76	2.8 ± 2.32	7.90 (<0.001)

* Korean won, tens of thousands; SD: Standard deviation

**Table 2 healthcare-11-02951-t002:** Differences in life satisfaction and social interaction by age group (n = 8188).

Variables		65–74 Age Group	≥75 Age Group	*t*(*p*)
		Mean ± SD		
Life satisfaction score (Range: 1–5)		3.6 ± 0.69	3.4 ± 0.71	14.23 (<0.001)
Social interaction (Range: 0–5)				
① Frequency of social activities				
Hobby clubs		0.1 ± 0.65	0.0 ± 0.40	8.13 (<0.001)
Social gatherings		1.1 ± 1.22	0.5 ± 0.93	28.69 (<0.001)
Religious activities		1.7 ± 1.88	1.5 ± 1.88	2.98 (0.003)
Voluntary activities		0.1 ± 0.50	0.0 ± 0.36	4.41 (<0.001)
Senior citizen community centers		0.8 ± 1.50	1.8 ± 2.01	−25.17 (<0.001)
② Frequency of social contacts				
Meeting	Children	2.5 ± 1.63	2.5 ± 1.23	−0.50 (0.619)
	Friends	4.0 ± 1.30	4.1 ± 1.43	−4.91 (<0.001)
Phone-call	Children	3.7 ± 1.27	3.7 ± 1.30	1.34 (0.818)
	Friends	4.0 ± 1.27	4.0 ± 1.49	1.68 (0.094)

S.D., standard deviation.

**Table 3 healthcare-11-02951-t003:** Social interaction factors affecting life satisfaction by age group.

Variables		Life Satisfaction					
		65–74 Age Group			≥75 Age Group		
		β	SE	*p*	β	SE	*p*
Social interaction							
① Frequency of social activities							
Hobby clubs		0.02	0.02	0.178	0.07	0.03	0.001
Social gatherings		0.03	0.01	0.074	0.02	0.02	0.470
Religious activities		0.01	0.01	0.567	0.06	0.01	0.063
Voluntary activities		0.04	0.02	0.007	−0.05	0.03	0.050
Senior citizen community centers		0.00	0.01	0.962	0.08	0.01	0.001
② Frequency of social contacts							
Meeting	Children	0.17	0.01	<0.001	0.01	0.02	0.722
	Friends	0.04	0.01	0.067	−0.02	0.02	0.465
Phone-call	Children	−0.03	0.01	0.099	0.07	0.02	0.013
	Friends	0.02	0.01	0.413	0.11	0.02	0.002
R^2^		0.251			0.267		
Adjusted R^2^		0.246			0.256		
F(*p*)		46.01 (<0.001)			23.22 (<0.001)		

Adjusted variables: sex, age, education level, religion, living arrangement, residential area, independence in daily help, perceived health status, perceived health status of spouse, housing ownership, annual household income, number of surviving children and grandchildren, and number of intimate kinships and friends.

## Data Availability

Data are contained within the article.

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
