# Peer review of "Social Interaction and Life Satisfaction among Older Adults by Age Group"

_healthcare, 2023, doi:10.3390/healthcare11222951_

Round 1
Reviewer 1 Report
Comments and Suggestions for Authors
In the text there is a motif of "silver halls". It is necessary to characterize these silver halls in a few sentences, so that the non-Korean reader will also become familiar with this specific form of support.
A sentence from the abstract: "In the prolonged old age of the elderly, knowledge and preparation for social interaction is essential to increase life satisfaction" is contextually incomprehensible to me.
Author Response
Response to Reviewer 1 Comments
Response 1
Thank you for your detailed review. Some sentences and words (e.g. silver halls, abstract sentence) were corrected. We once again received professional editing services for English language. We think the language related issues have been improved. The editing certificate is attached. We also revised many parts of the paper. Your comment was very helpful. Thank you for your time and effort.

Reviewer 2 Report
Comments and Suggestions for Authors
This is an interesting study. It has the potential to contribute to our cross-cultural understanding of aging and life satisfaction. My comments are included below.
1. I would like to know the extent to which individuals who are 65 or older, and live individually or with their spouse in Korea.
2. I would like to see more theoretical or empirical justifications for why the authors divided the sample into two age groups: 65-75 and 75>. Without justifications, this division seems to be very arbitrary.
3. The results reported in Table 2 are informative. However, the multivariate regression analysis can be improved. I encourage the authors to do the following:
(1) Create an index variable of social activities by selecting significant items from Table 2 (this can be a sum-score or mean-score index variable). Please report the reliability coefficient.
(2) Similarly, create an index variable of social relationships by combining the visitation items from Table 2 (this can be a sum-score or mean-score index variable). Please report the reliability coefficient.
(3) By the same token, create an index variable of social relationships by combining the phone-call items from Table 2 (this can be a sum-score or mean-score index variable). Please report the reliability coefficient.
(4) I would reconsider the satisfaction items (with children and friends) that can be perceived as the dimensions of life satisfaction.
(5) I recommend the following regression models: Model 1 is the social activity model with statistical controls for both age groups; Model 2 is the social relationship model using the frequency of visits with statistical controls for both age groups; Model 3 is the social relationship model using the frequency of phone-calls with statistical controls for both age groups; and finally if the authors are interested in the age group differences or similarities, statistical tests of the regression coefficients across the two age groups can be conducted.
Comments on the Quality of English Language
Some of the paragraphs are two short. the manuscript will red better if it is line-edited by a native speaker who does aging research.
Author Response
Response to Reviewer 2 Comments
Response 2
Thank you for your comprehensive review of our manuscript. We have carefully examined all the comments by you and the reviewers in the manuscript and agree with them.
1.The title was modified, and the participants for whom the data was analyzed were additionally described to reduce reader confusion in the methods section.
2. Based on previous studies, we added evidence to divide the elderly into early life and late life. Neugarten, B., Moom, J., & Low, L. (1995). Age constraints and adult socialization. American Journal of Sociology, 70, 710-717.
3. Thank you very much for your thoughtful thoughts on statistics. Statistical analysis was conducted based on reviewer opinions. However, because the reliability of the tool was not significant, it could not proceed as is. So, after various analysis results, the model was verified by treating the main variables as continuous variables. Thank you for making the results of the paper clear. We also revised many parts of your concern in the paper. Your comment was very helpful. Thank you for your time and effort.
4. The overall content of the paper was strengthened. We once again received professional editing services for English language. We think the language related issues have been improved. The editing certificate is attached.

Reviewer 3 Report
Comments and Suggestions for Authors
In this study, the authors investigate the effect of social interaction on life satisfaction. They divide respondents into two categories based on age and compare their life satisfaction based on social activities, social relationships, and satisfaction with relationships with children and friends. I agree that this is important work, determining how older adults feel satisfaction in their lives and what factors can influence this. I do, however, have some concerns with the manuscript, primarily that the literature review does not adequately illustrate the current research in the field and it is not clear how the findings are novel and contribute to a larger theoretical question.
Introduction
· Lines 48-50: Can you more explicitly connect the deterioration of physical functions to the decline of social interactions?
· Line 51: What do you mean that maintenance of social interactions may surpass life stages?
· Line 52: The growth of older adults living in single and couple households needs a citation.
· Lines 53-56: This also needs more explanation. Aren’t single people and coupled people all people? That makes the next subject, “these individuals” confusing. You might reframe this to be people living with or without adult children are different. Are people living with partners experiencing social isolation and loneliness?
· Given this article is about life satisfaction related to social interactions, I would expect to see a deeper discussion on the literature in this area. Currently, there is a brief mention about social interaction and happiness (Lines 58-60), within a list of factors that impact satisfaction. A broader and deeper dissection of this literature is needed.
· Line 71: Again, I’m confused by the “single-person households and couples” distinction. Is the idea to compare people who live alone and people who live with others? If so, should that not be the distinction made? Is it about having a romantic attachment? Is “couples” really measuring married people? Couples can also be people who live together unmarried or are partnered but not cohabiting.
· Line 73-74: This makes it sound like the article is a literature review. You can remove this sentence (and edit the beginning of the next sentence to be clear you are talking about what this study accomplishes).
· Line 87: This feels redundant given point #2 notes you’re looking at differences by age group. I recommend dropping this point or removing “by age group” from point #2.
Materials and Methods
· Line 94: It is worth explaining why you left out people who were helped in completing the survey by children and others. If you’re looking at social life satisfaction between those who live alone and those who are partnered, wouldn’t excluding people who have close enough relationships with others to the point that they rely on them for help skew the results? This may not be the case, but it is worth justifying this design decision.
o Where results any different if those individuals remained in the sample?
· Line 119: What are silver halls?
· Line 121: How often is “regular” use of facilities?
· Line 124: “Family refers to family relationships” is not descriptive. Who is included in “family?” Is this immediate family, extended family, friends so close they are perceived as family?
· Line 125: Here also, the definition of friends is vague. The sentence, “Friends include friends and neighbors as close and friendly people,” is unclear, does not give a good idea of how “friends” are measured, or who is included.
· Line 149: What does it mean that variables were adjusted?
· I appreciate the inclusion of the Ethical Considerations!
Results
· In the second paragraph, there are a few cases where dissimilar characteristics are listed in the same sentence, making them seem like they go together, awkwardly. This is the case for education and religion, living in a couple and rural living, and living independently and health status. There were some similar sentence structure issues in the following paragraph.
· Using an asterisk * to note in the table which results are statistically significant would be helpful.
· Please clarify the reference group. Were results comparing participation or not participating in certain activities, or (not) participating compared to the other age group?
· Line 237: This is unnecessary. You can simply state your findings, as you do in the next paragraph.
Discussion
· Paragraph starting at line 280: This paragraph needs to be reorganized. The paragraph starts with SST, but does not describe what SST is or how it applies. The remaining parts of the paragraph are unrelated to SST, which focuses on how the number and types of relationships change with age, as one focuses more on rewarding relationships and less on the less-rewarding relationships. Additionally, if SST is going to be a central part of the Discussion, it would be worth including this and possibly other theories in the literature review.
· Starting at line 281 and for the next few paragraphs, the discussion on physical aging and decline feels quite out of place. At no point in the manuscript prior was physical health connected to social interaction. If the connection between physical health and social interaction or satisfaction is central to the argument, this needs more a more explicit connection and a more thorough relationship to the findings on getting together with friends and family. Was there anything in the data that illustrated those in worse health had less interaction with others or were not as satisfied with social relationships?
· Given how the literature review was light on prior studies’ findings on social relationships in later life, it is unclear what novel findings this research contributes. It seems well-known that social interactions are important to life satisfaction and overall well-being. The Discussion needs to more clearly articulate what of this study is a new finding and how that impacts the larger conversation.
Conclusion
· Line 319: Again, the relationship between physical aging and social interactions needs to be clear. This feels unconnected to the results.
Comments on the Quality of English LanguageI had no concerns with the quality of English language in the manuscript. There were a few places where sentence structure was slightly hard to follow, but more in a writing style than an issue with English language.
Author Response
1.Thank you for your comprehensive review of our manuscript. We have carefully examined all the comments by you and the reviewers in the manuscript and agree with them.
2.We read the reviewer comments carefully. After reading your reviewer comments, we understand your concerns. We comprehensively revised the entire paper according to the comments.
3.Statistical analysis was also performed again to clearly organize the results of the paper. We did our best to make corrections so that your hard work was not in vain.
4.We revised the paper title-introduction-method-results-discussion.
5.We also received professional editing service one more time. Thank you for giving us the opportunity to upgrade our thesis once again.
6.Your comment was very helpful. Thank you for your time and effort.

Round 2
Reviewer 2 Report
Comments and Suggestions for Authors
The revisions directly addressed the reviewer's concerns/comments. The analysis and the presentation of the results can now be followed without difficulties. However, the quality of the presentation needs a bit more work. I will say this again: small paragraphs should be combined if possible. There are many short paragraphs with two sentences, which affect the paper's readability.
Comments on the Quality of English LanguageAnother round of extensive English line editing is needed.
Author Response
Response 2
We are very grateful for your concern and interest in our paper. Your comments have greatly improved the quality of our paper. In particular, we spent a lot of time revising the statistical analysis, and as a result, it grew into a good paper.
Once again, based on your comments, we've combined the smaller paragraphs into one paragraph and improved readability. Your detailed instructions helped us even more.
We would like to express our gratitude to the reviewers and healthcare officials who gave us the opportunity to make corrections.
